# Obstructive Sleep Apnea and Atrial Fibrillation

**DOI:** 10.3390/jcm11051242

**Published:** 2022-02-25

**Authors:** Amalia Ioanna Moula, Iris Parrini, Cecilia Tetta, Fabiana Lucà, Gianmarco Parise, Carmelo Massimiliano Rao, Emanuela Mauro, Orlando Parise, Francesco Matteucci, Michele Massimo Gulizia, Mark La Meir, Sandro Gelsomino

**Affiliations:** 1Cardiothoracic Department, Maastricht University Hospital, 6229 HX Maastricht, The Netherlands; amaliamoula1@gmail.com (A.I.M.); cecilia.tetta@ior.it (C.T.); g.parise@maastrichtuniversity.nl (G.P.); emanuelamauroam@gmail.com (E.M.); o.parise@icloud.com (O.P.); melampo6@gmail.com (F.M.); lameir@yahoo.com (M.L.M.); 2Cardiology Department, Mauriziano Umberto I Hospital, 10128 Torino, Italy; irisparrini@libero.it (I.P.); massimo.rao@libero.it (C.M.R.); 3Cardiology Department, Big Metropolitan Hospital, 89129 Reggio Calabria, Italy; fabiana.luca92@gmail.com; 4Cardiology Department, Garibaldi Nesima Hospital, 95122 Catania, Italy; michele.gulizia60@gmail.com; 5Heart Care Foundation, 50121 Firenze, Italy; 6Cardiothoracic Department, Brussels University Hospital, 1099 Jette, Belgium

**Keywords:** atrial fibrillation, arrhythmia, obstructive sleep apnea

## Abstract

Atrial fibrillation (AF) is the most common arrhythmia, increasing with age and comorbidities. Obstructive sleep apnea (OSA) is a chronic sleep disorder more common in older men. It has been shown that OSA is linked to AF. Nonetheless, the prevalence of OSA in patients with AF remains unknown because OSA is significantly underdiagnosed. This review, including 54,271 patients, carried out a meta-analysis to investigate the association between OSA and AF. We also performed a meta-regression to explore cofactors influencing this correlation. A strong link was found between these two disorders. The incidence of AF is 88% higher in patients with OSA. Age and hypertension independently strengthened this association, indicating that OSA treatment could help reduce AF recurrence. Further research is needed to confirm these findings. Atrial Fibrillation (AF) is the most common arrhythmia, increasing with age and comorbidities. Obstructive sleep apnea (OSA) is a regulatory respiratory disorder of partial or complete collapse of the upper airways during sleep leading to recurrent pauses in breathing. OSA is more common in older men. Evidence exists that OSA is linked to AF. Nonetheless, the prevalence of OSA in patients with AF remains unknown because OSA is underdiagnosed. In order to investigate the incidence of AF in OSA patients, we carried out a meta-analysis including 20 scientific studies with a total of 54,271 subjects. AF was present in 4801 patients of whom 2203 (45.9%) had OSA and 2598 (54.1%) did not. Of a total of 21,074 patients with OSA, 2203 (10.5%) had AF and 18,871 (89.5%) did not. The incidence of AF was 88% higher in patients with OSA. We performed a meta-regression to explore interacting factors potentially influencing the occurrence of AF in OSA. Older age and hypertension independently strengthened this association. The clinical significance of our results is that patients with OSA should be referred early to the cardiologist. Further research is needed for the definition of the mechanisms of association between AF and OSA.

## 1. Introduction

Atrial fibrillation (AF) is the most common arrhythmia, with high mortality and morbidity [1,2,3]. Its incidence and the social and economic burden of AF on healthcare systems worldwide are rising significantly [1,2,4,5,6,7]. Unfortunately, despite extensive research, the mechanisms underlying AF are complex and not completely understood yet [1,2,8,9,10,11,12,13,14,15,16,17,18]. Obstructive sleep apnea (OSA) is a common chronic disorder affecting about 2–4% of the adult population, being more common in old men [19]. The condition is characterized by repetitive episodes of the complete or partial collapse of the upper airway during sleep, with a consequent cessation/reduction of the airflow [20]. Sleep apnea has been implicated in the pathogenesis of multiple cardiovascular diseases (CVD), including arrhythmias, hypertension, heart failure (HF), and stroke [21,22]. A link between AF and OSA [23,24,25,26] has been described, and it has been assessed that these two pathological conditions share many common unmodifiable and modifiable risk factors, including sex, age, obesity, diabetes mellitus, smoking, *Helicobacter pylori* infection, etc. [3,14,27,28,29].

Nonetheless, the prevalence of OSA in patients with AF is unknown because OSA is generally underdiagnosed [19,20]. Furthermore, controversy over this association and its directionality exists, because of the high incidence of cardiovascular (CV) comorbidities in patients with OSA and AF. Patients with OSA have a higher incidence of AF than the general population [23,25,30].

We performed a review involving 54,271 patients, and we assessed the association between OSA and AF. In addition, we carried out a meta-regression to test whether this association was independent of other shared cardiovascular risks.

## 2. Materials and Methods

### 2.1. Search Strategy

The literature search was performed in agreement with the principles of the Preferred Reporting Items for Systematic Reviews and Meta-Analyses (PRISMA) [31] and the Cochrane handbook [32].

An unrestricted literature search was performed using PubMed, Web of Science and Google Scholar Databases. The PubMed Database was selected as the main database to perform this search.

The used PubMed search items were the following: (“Sleep Apnea Syndromes” [Mesh] OR “sleep apnea”) AND (“atrial fibrillation” [Mesh] OR “atrial fibrillation”). Only articles written in English were examined.

The search strategy was decided by two authors (IP and CT), and a third author (MLM) approved the decisions. The literature search was performed by one author (AIM), the selected articles’ eligibility, and the risk of bias was assessed independently by two reviewers (FL and MMG). The assessment of bias at the individual study level was done using the Risk of Bias in Non-randomized Studies-of Interventions (ROBINS-I) tool [33]. The included studies were independently assessed for the risk of bias at the individual study level by two reviewers (AIM and GP). After discussion, a third reviewer (SG) was involved in resolving possible disagreements. The risk of bias was assessed for the following causes: (1) in the confounding; (2) in the selection of participants into the study; (3) in the classification of interventions; (4) due to deviations from intended interventions; (5) due to missing data; (6) in the measurement of outcomes; (7) in the selection of the reported result; and (8) overall bias assessment. The Cochrane Handbook was used for the evaluation of the examined domains for bias. The Risk-of-Bias VISualization (robvis) software was used for the production of the plot for ROBINS-I [34].

### 2.2. Selection Process

The article selection was based on defined inclusion criteria. These criteria were the following: (1) human studies; (2) full articles about AF and OSA having a non-AF control population; (3) studies containing adequate information regarding the presence of OSA and AF and (4) studies including at least 10 patients.

The exclusion criteria for the article selection were: (1) non-human studies, (2) case reports, (3) previous reviews and/or meta-analyses, (4) editorials, (5) studies reporting AF status after continuous positive airway pressure (CPAP) therapy, (6) studies reporting AF and/or OSA status after ablation/other interventional AF treatment, and (7) studies without data regarding both the AF and OSA status of the included patients.

### 2.3. Quality Assessment 

The quality of included studies was assessed using a rating scale based on the Downs and Black checklist for measuring [35]. A version including 18 items was employed, using a binary score (0 or 1) except for two items which are rated on a scale from 0 to 2 and from 0 to 5, respectively.

Two independent researchers (CT and GP) collected the ratings. Any divergences were resolved by a third reviewer (OP) and quantified using Cohen’s kappa [36].

### 2.4. Definitions 

AF was defined as supraventricular tachyarrhythmia with uncoordinated atrial activation and consequently ineffective atrial contraction as diagnosed in an electrocardiogram by irregular R-R intervals when atrioventricular (A-V) conduction is present, absence of distinct repeating P waves and irregular atrial activity [2]. OSA was defined as a disorder characterized by frequent stops of breathing during sleep resulting from obstruction of the upper airway that occurs because of the inadequate motor tone of the tongue and/or airway dilator muscles [37].

### 2.5. Statistical Analysis

The meta-analysis was conducted using v. 3.6.1 (R Foundation for Statistical Computing, Vienna, Austria). The log of the odds ratios (Log OR) was used as index statistics. The random effects model was employed because heterogeneity among studies was anticipated. Heterogeneity was evaluated with the statistical inconsistency Higgin’s I2 test [38]. I2 values < 40% were considered having low heterogeneity, I2 values > 75% were considered having high heterogeneity. Publication bias was assessed using Egger’s test of the intercept. In addition, a meta-regression analysis has been performed to explore the impact of the following potential interaction factors: sex, age, diabetes mellitus, hypertension, and BMI on the occurrence of AF in OSA patients. *p* values < 0.05 were considered statistically significant.

## 3. Results 

### 3.1. Search Results and Characteristics of the Studies

The initial search produced 768 results. After the application of the inclusion and exclusion criteria, 32 articles that included data on the simultaneous presence of AF and OSA were found. After rejecting articles without separate data for patients with AF and OSA, a total of 15 articles were found to fulfill the criteria. After conducting a free search and searching the references of the selected articles, five additional papers were added (Figure 1). Finally, the selection process resulted in 20 studies that were included in the analysis [30,39,40,41,42,43,44,45,46,47,48,49,50,51,52,53,54,55,56,57].

The selected papers included a total of 54,271 subjects. AF was present in 4801 patients of which 2203 (45.9%) had OSA and 2598 (54.1%) did not. Of a total of 21,074 patients with OSA, 2203 (10.5%) had AF and 18,871 (89.5%) did not show AF. The studies which were included in this meta-analysis and the patients’ characteristics are shown in Table 1.

### 3.2. Quality of the Studies

The quality assessment is shown in Appendix A. The median rating was 0.78 [IQR 0.36–0.84], ranging from 0.44 to 1.6. Acceptable inter-rater agreement was found (κ = 0.9; agreement = 92.3%).

### 3.3. Relation between OSA and AF

The composite log odds ratio for all the studies was 0.63 (*p* < 0.001), indicating a strong positive link between the two diseases. More specifically, the balance between AF+ and AF− in patients with OSA is 1.88 greater than this ratio in patients without OSA. In other words, the incidence of AF is 88% higher in patients with OSA. The log of the odds ratios (Log OR) for supraventricular arrhythmias and OSA (Figure 2A) and AF and OSA (Figure 2B) are shown in Figure 2 (funnel diagram in Figure 3). There was no significant difference between those papers that refer to supraventricular arrhythmias and those that refer to AF (0.63 [0.44, 0.83] vs. 0.68 [0.48, 0.89], respectively).

The results were further analyzed for the possible effects of sex, age, diabetes mellitus, hypertension, and BMI. The respective diagrams are shown in Figure 4, Figure 5, Figure 6, Figure 7 and Figure 8. Age (*p* = 0.04) and hypertension (*p* = 0.04) were found to affect the incidence of OSA in AF patients. An increase of 5 years corresponded to a reduction by 9.4% in incidence of AF. In contrast, no effect was found for sex (0.44), diabetes mellitus (*p* = 0.62), and BMI (*p* = 0.9).

### 3.4. Discussion

A significant association between OSA and AF has been advocated [58] as well as a significant “interplay” between these two pathological entities [58]. A recent scientific statement from the American Heart Association (AHA) has confirmed this association [59].

In contrast, the prevalence of central sleep apnea (CSA) in patients with AF is less clarified, although it is high in patients with HF and reduced left ventricular ejection fraction (LVEF) [60].

However, OSA and AF share many common risk factors [61]. Therefore, the association of OSA and AF might be due to shared CVD and obesity. Hence, a direct link between the two diseases is still debated since the close association between CVD and both these pathologies, might obscure a directly causal relationship between OSA and AF. We performed a review of the published paper, involving 54,271 patients assessing the incidence of AF in OSA patients. We found a strong association between OSA and AF. Indeed, we found that the incidence of AF is 88% higher in patients with OSA.

This confirms the results from Chao et al. including 579,521 patients, of which 4082 had OSA. After a 9.2-year average follow-up, the incidence of AF was 0.7% in patients without OSA and 1.38% in patients with OSA (*p* < 0.001) [62]. Nonetheless, OSA and AF share many common risk factors therefore, the presence of one may promote the development of the other. Hence it is not clear whether this association is primary or mediated by shared risk factors. For this reason, we wanted to go further, carrying out a meta-regression to test whether this association was independent, or it was the results of the interaction of OSA with other shared cardiovascular risks with common pathophysiological mechanisms. The risk factors that the two diseases have include: Obesity or high body max index (BMI), older age, hypertension, diabetes, smoking, dyslipidemia, and male gender [61,63,64,65]. 

Of the abovementioned factors, age and hypertension were found in this meta-analysis to significantly contribute to the correlation between AF and OSA. These factors are independently related to both AF and OSA. Additionally, these risk factors are also closely interrelated. Hypertension results from multiple genetic and environmental factors, however, it is closely related to the aging process and the resulting increasing arterial stiffness [66].

Increased age was found to be correlated with 9.4% reduced incidence of AF in patients with OSA (*p* < 0.05). Although children and adolescents can also have OSA, the prevalence of the disease is significantly higher in adults and increases with age [67]. The prevalence of AF is also associated with aging. Therefore, the mechanisms involved into such a reduction in AF incidence are not entirely known.

The second factor that was found to be correlated with increased incidence of AF in patients with OSA was hypertension that contributes to the development of anatomical abnormalities that lead to the development of AF [68].

A recent meta-analysis showed that daytime, nighttime, and 24-h systolic blood pressure (BP) are similarly associated with future AF and that ambulatory systolic BP is a better predictor than clinic BP [69]. The mechanism involved in the effect of hypertension on AF is not well known but two main mechanisms are proposed. First, hypertension can cause hemodynamic changes and an increase of pressure in the left atrium leading to enlargement of the atrium, and second, hypertension-related activation of the renin-angiotensin-aldosterone system (RAAS) and autonomic dysregulation can also induce fibrosis in the left atrium (LA) contributing to the development of AF [70].

Hypertension and OSA have several common factors in pathophysiology, including gender, obesity, lifestyle, impaired quality of sleep, RAAS, and increased fluid distribution [63]. OSA is a cause of secondary hypertension as apneas during the night can result in surges in blood pressure and subsequently elevated average systolic and diastolic pressure.

The other factors that were examined, namely BMI, diabetes, smoking, dyslipidemia, and sex were not found to influence the correlation between OSA and AF even though they are known to be predictors for both OSA and AF.

Although these connections are still speculative, there are possible pathophysiological links between AF and OSA. Having OSA may lead to initiation or exaggeration of the effects of AF through a series of mechanisms [71]. OSA patients, can develop negative intrathoracic pressure [72]. This negative pressure may lead to an increase of afterload as well as atrial stretch [61]. As a result, there can be left ventricular hypertrophy (LVH) and electrical remodeling leading to increased chances of AF [61]. OSA may also lead to smaller refractory and action potential periods that arise from the increased vagal and sympathetic activity leading to structural and electrical remodeling in the atria that contribute to the onset of AF [73]. Furthermore, a rise in CO_2_ and reduction of O_2_, leads to an increase in inflammation and oxidative stress that may also contribute to the development of AF [74]. So, OSA may lead to initiation or worsening of AF via the distortion of the normal electrophysiological mechanisms of the heart [26,58,75,76]. Under this perspective it can be hypothesized that treatment of OSA may contribute to reducing the AF symptoms and this hypothesis is supported by research data. In a recent meta-analysis, it was interestingly shown that CPAP treatment of OSA is associated with a reduction of AF recurrence in patients who were not treated with radiofrequency ablation (RFA) or electrical cardioversion(ECV) [77]. In addition, non-selective B blockers (BB) may reduce heart rate (HR) during OSA and avoid bradycardias.

Our results clearly quantify the connection between these two diseases and that patients with OSA have a higher risk for developing AF. This can have major implications in the treatment of patients who have both disorders. Other studies also suggest that treating OSA could lead to better outcomes in co-existing AF.

### 3.5. Limitations

There are some limitations in this study that need to be pointed out:

First, the definitions of AF and OSA and the respective diagnostic methods are not universal and there were partial differences between the studies. Some studies defined OSA based on the apnea-hypopnea index (AHI) and other based on the respiratory disturbance index (RDI). There is a possibility that differences may occur from this difference [60]. Second, two of the included studies [41,44] were retrospective. Third, some of the selected articles included specific categories of patients or patients with specific characteristics. One study included only diabetic patients [39] and another one included patients scheduled to undergo cardiac surgery [42]. Fourth, not all the included studies had detailed data about the characteristics, such as age, sex, BMI, and presence of diabetes or hypertension for all the subgroups of patients (i.e., with and without AF and OSA). Fifth, day-to-day variability is a weak point in diagnostics of either the FA runs or the OSA events. It should be considered and adjusted for since it might be a source of underdiagnosing of the mentioned clinical conditions. Unfortunately, all the AF and OA studies share the limitation of using hazard functions for events that cannot meet the hazard assumption and whose initiation and termination cannot be clearly defined. When the continuous implantable monitoring will be employed in OSA patients with AF more insights can be obtained. Finally, compared to control subjects, men with AF seem to have thicker necks, and patients with lone AF report more daytime tiredness, daytime sleepiness, and breathing pauses during sleep This has to be taken into account while looking at our results [51].

## 4. Conclusions

Our results show a strong correlation of the existence of OSA with the risk for AF. Older age and hypertension strengthen this correlation. This suggests that patients with OSA should be referred to a cardiologist for a strict follow up including rest ECG and Holter ECG. Further research is needed for the definition of the mechanisms of association between AF and OSA.

## Figures and Tables

**Figure 1 jcm-11-01242-f001:**
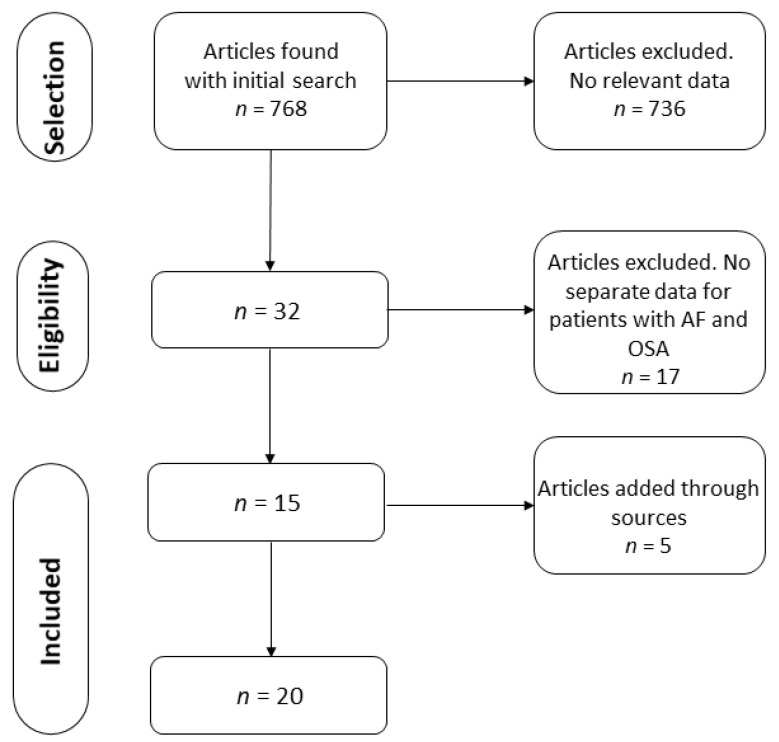
PRISMA flow chart of the article selection process. AF = Atrial fibrillation. OSA = Obstructive Sleep Apnea.

**Figure 2 jcm-11-01242-f002:**
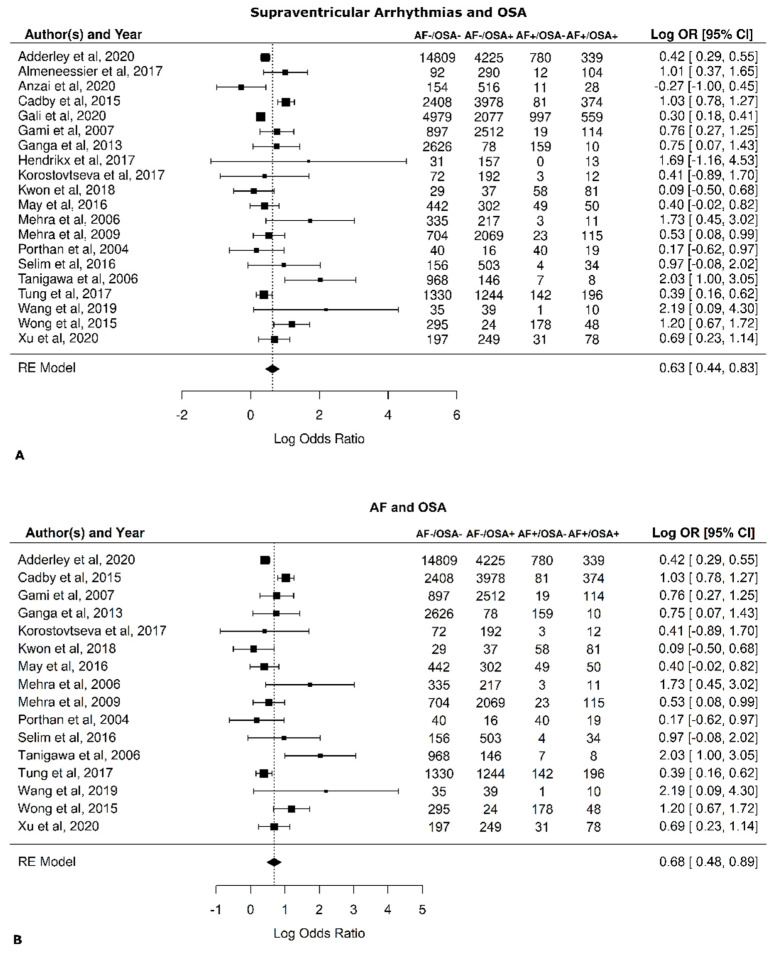
Risk of atrial fibrillation (AF) in patients with obstructive sleep apnea (OSA). Forest plots. (**A**) All references, including those that did not discriminate between arrhythmia in general and AF, were examined [30,39,40,41,42,43,44,45,46,47,48,49,50,51,52,53,54,55,56,57]. (**B**) Only references that include only AF were examined [39,41,43,44,46,47,48,49,50,51,52,53,54,55,56,57].

**Figure 3 jcm-11-01242-f003:**
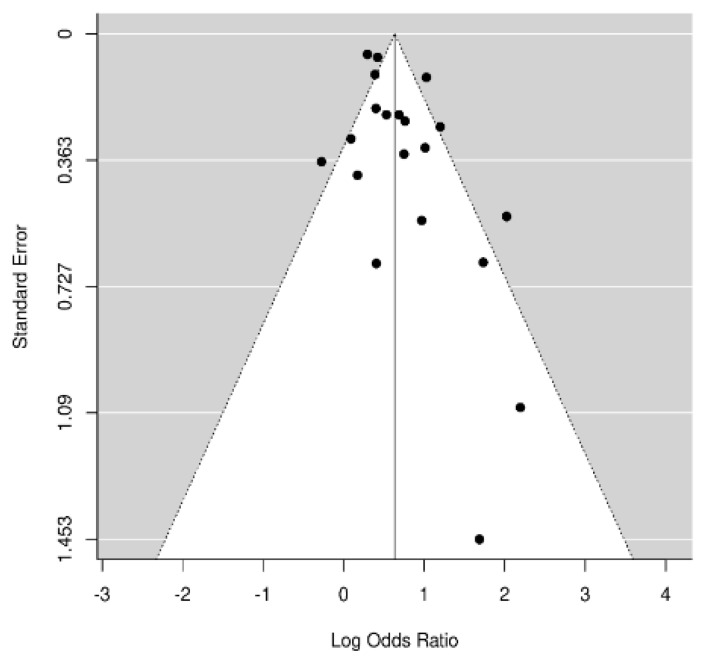
Risk of atrial fibrillation (AF) in patients with obstructive sleep apnea (OSA). Funnel plot.

**Figure 4 jcm-11-01242-f004:**
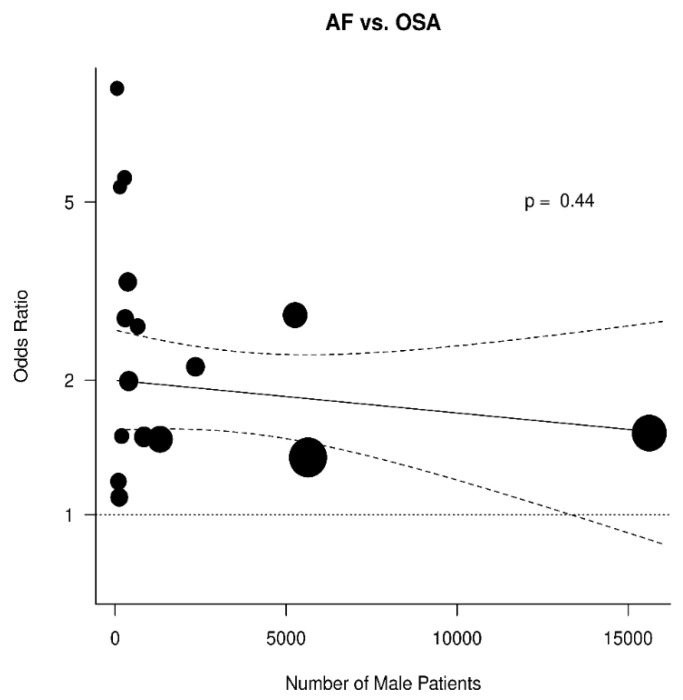
Effect of sex in the odds ratio for atrial fibrillation (AF) in patients with obstructive sleep apnea (OSA).

**Figure 5 jcm-11-01242-f005:**
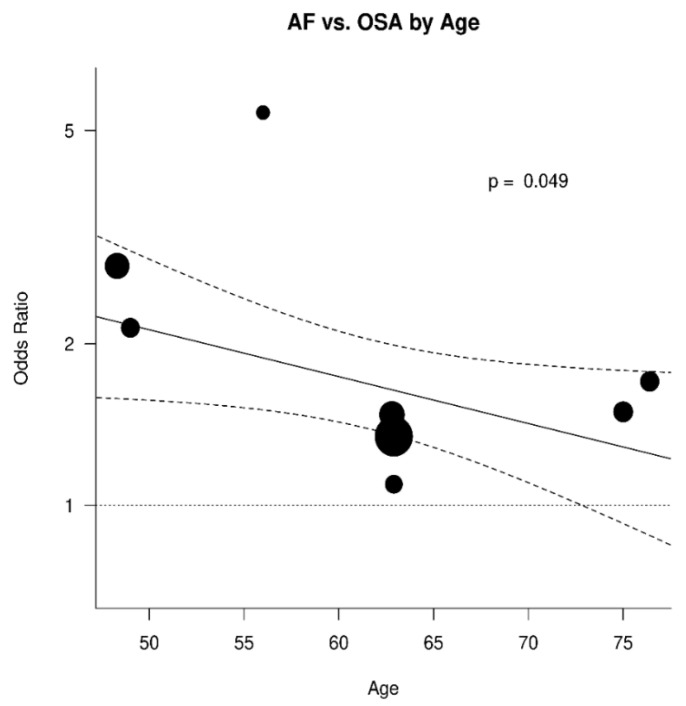
Effect of age in the odds ratio for atrial fibrillation (AF) in patients with obstructive sleep apnea (OSA).

**Figure 6 jcm-11-01242-f006:**
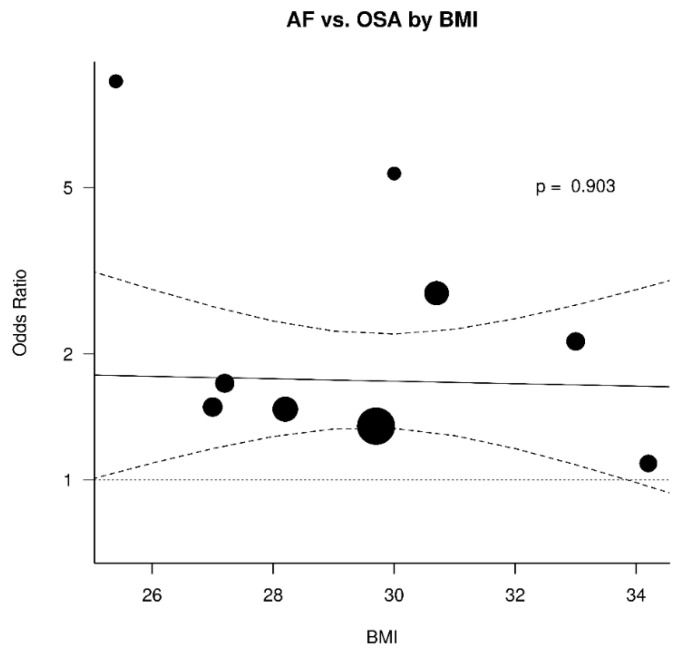
Effect of body mass index (BMI) in the odds ratio for atrial fibrillation (AF) in patients with obstructive sleep apnea (OSA).

**Figure 7 jcm-11-01242-f007:**
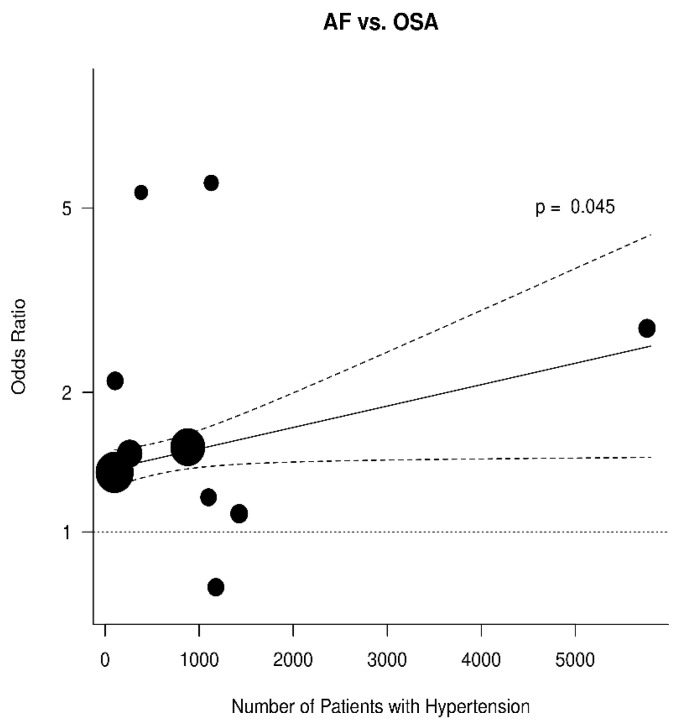
Effect of hypertension in the odds ratio for atrial fibrillation (AF) in patients with obstructive sleep apnea (OSA).

**Figure 8 jcm-11-01242-f008:**
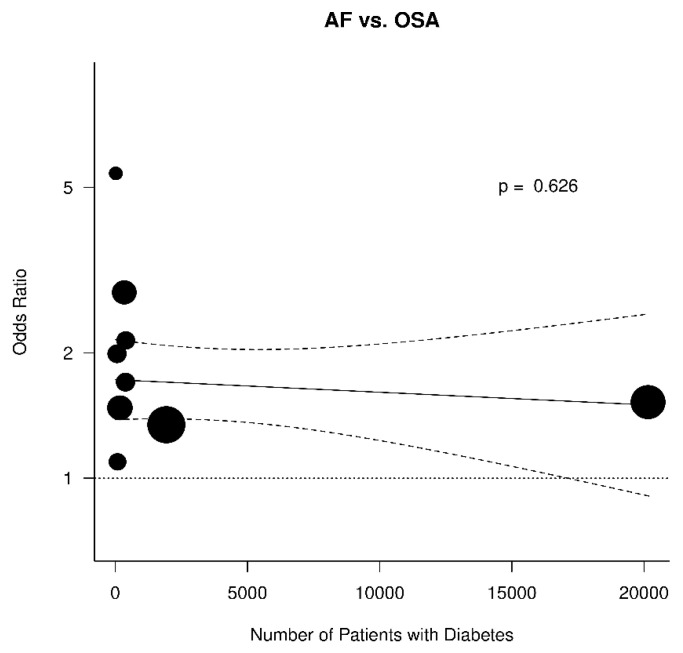
Effect of diabetes mellitus in the odds ratio for atrial fibrillation (AF) in patients with obstructive sleep apnea (OSA).

**Table 1 jcm-11-01242-t001:** Patients’ characteristics.

Paper #	Paper	Total Patients	Total AF+	AF+ OSA+	AF+ OSA−	Total AF−	AF− OSA+	AF− OSA−	Total OSA+	Total OSA−
1	Adderley et al., 2020 [39]	20,153	1119	339	780	19,034	4225	14,809	4564	15,589
2	Almeneessier et al., 2017 [30]	498	116	104	12	382	290	92	394	104
3	Anzai et al., 2020 [40]	709	39	28	11	670	516	154	544	165
4	Cadby et al., 2015 [41]	6841	455	374	81	6386	3978	2408	4352	2489
5	Gali et al., 2020 [42]	8612	1556	559	997	7056	2077	4979	2636	5976
6	Gami et al., 2007 [43]	3542	133	114	19	3409	2512	897	2626	916
7	Ganga et al., 2013 [44]	2873	169	10	159	2704	78	2626	88	2785
8	Hendrikx et al., 2017 [45]	201	13	13	0	188	157	31	170	31
9	Korostovtseva et al., 2017 [46]	279	15	12	3	264	192	72	204	75
10	Kwon et al., 2018 [47]	205	139	81	58	66	37	29	118	87
11	May et al., 2016 [48]	843	99	50	49	744	302	442	352	491
12	Mehra et al., 2006 [49]	566	14	11	3	552	217	335	228	338
13	Mehra et al., 2009 [50]	2911	138	115	23	2773	2069	704	2184	727
14	Porthan et al., 2004 [51]	115	59	19	40	56	16	40	35	80
15	Selim et al., 2016 [52]	697	38	34	4	659	503	156	537	160
16	Tanigawa et al., 2006 [53]	1129	15	8	7	1114	146	968	154	975
17	Tung et al., 2017 [54]	2912	338	196	142	2574	1244	1330	1440	1472
18	Wang et al., 2019 [55]	85	11	10	1	74	39	35	49	36
19	Wong et al., 2015 [56]	545	226	48	178	319	24	295	72	473
20	Xu et al. 2020 [57]	555	109	78	31	446	249	197	327	228

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
