# Peer review of "Obstructive Sleep Apnea and Atrial Fibrillation"

_jcm, 2022, doi:10.3390/jcm11051242_

Round 1
Reviewer 1 Report
The paper covers an important subject. The methodology seems adequate, but i could not see the figures. Please add them.
Author Response
1_Reviewer Comments and Suggestions for Authors
The paper covers an important subject. The methodology seems adequate, but i could not see the figures. Please add them.
Answer:
Dear Reviewer: Thank you for your insightful comment. We agree with this comment and the figures have been added at the end of the manuscript.
Reviewer 2 Report
Review_JCM_1413565
I have read thoroughly your manuscript and there are several issued to be addressed, namely:
Title: why is „running head“ incorporated in the title of the manuscript?
Abstract:
Incohorent with very low informational value. This 54, 271 patients were primarily diagnosed with OSA or AF, or BOTH conditions? (You have to state exactly) That is a huge difference? Moreovere that is demanding to find not cofactor but CONFOUNDERS, since if you have 54 thousands of patient, then every correlation will be significant in terms of p value!
Statistical analysis
Have you considered a funnel test?Which would hipefully cover more than publiction bias
Relation between OSA and AF
U state that age has a relation to OSA, to what extent, may you plese quantify wether there is some cut-off value at which the age is considered higher risk factor than the other mentioned?
Gender
You stated that no effect was found for sex, fro the antropometric point of view that is surprising. First men have are much more often affected and on the other hand female sex is a risk factor even in the CHADSVASc score. You should mention what was the proportion of men vs. women in the whole cohort and than analyze separately for both genders.
Moreover a baseline characteristics including age, antropometric parameters (BMI, height, weight), prevalence of hypertension, coronary artery disease etc.
Conclusion
The conclusion is completely wrong. You CANNOT assume that treating OSA will reduce the incidence of AF! What data have you shown to draw such a strong conclusion? Indeed the result part accounts for a minority of the article.
Author Response
I have read thoroughly your manuscript and there are several issued to be addressed, namely:
- Title: why is „running head“ incorporated in the title of the manuscript?
Answer:
Dear reviewer, thank you for your comment. The “running head” was removed accordingly.
- Abstract:
Incohorent with very low informational value. This 54, 271 patients were primarily diagnosed with OSA or AF, or BOTH conditions? (You have to state exactly) That is a huge difference? Moreovere that is demanding to find not cofactor but CONFOUNDERS, since if you have 54 thousands of patient, then every correlation will be significant in terms of p value!
Answer:
Dear reviewer, thank you for your comment. We agree that the aim of the manuscript was not stated clearly, that was to study the true incidence of AF in patients with OSA. Indeed, in our clinical practice we are inviting doctors to refer these patients to the cardiologist as soon as possible and this is the main message of the paper. Regarding the statistical issue, this reviewer is completely right because we meant to say that an interaction of cardiovascular risk factors was studied to explore whether this association AF and OSA was independent or strengthened by interaction with these risk factors.
Original abstract: Atrial Fibrillation (AF) is the most common arrhythmia, increasing with age and comorbidities. Obstructive sleep apnea (OSA) is a chronic sleep disorder more common in older men. It has been shown that OSA is linked to AF. Nonetheless, the prevalence of OSA in patients with AF remains unknown because OSA is significantly underdiagnosed. This review, including 54,271 patients, carried out a meta-analysis to investigate the association between OSA and AF. We also performed a meta-regression to explore cofactors influencing this correlation. A strong link was found between these two disorders. The incidence of AF is 88% higher in patients with OSA. Age and hypertension independently strengthened this association, indicating that OSA treatment could help reduce AF recurrence. Further research is needed to confirm these findings.
Changes:
Atrial Fibrillation (AF) is the most common arrhythmia, increasing with age and comorbidities. Obstructive sleep apnea (OSA) is a regulatory respiratory disorder of partial or complete collapse of the upper airways during sleep leading to recurrent pauses in breathing. OSA is more common in older men. Evidence exists that OSA is linked to AF. Nonetheless, the prevalence of OSA in patients with AF remains unknown because OSA is underdiagnosed. In order to investigate the incidence of AF in OSA patients, we carried out a meta-analysis including 20 scientific studies with a total of 54,271 subjects. AF was present in 4,801 patients of whom 2,203 (45.9%) had OSA and 2,598 (54.1%) did not. Of a total of 21,074 patients with OSA, 2,203 (10.5%) had AF and 18,871 (89.5%) did not. The incidence of AF was 88% higher in patients with OSA. We performed a meta-regression to explore interacting factors potentially influencing the occurrence of AF in OSA. Older age and hypertension independently strengthened this association. The clinical significance of our results is that patients with OSA should be referred early to the cardiologist. Further research is needed for the definition of the mechanisms of association between AF and OSA.
- Statistical analysis
Have you considered a funnel test? Which would hopefully cover more than publication bias
Answer:
Dear reviewer. Thank you for your remark. A funnel was included in the original submission as part of figure 2. The original legend of figure 2 is: “Forest plot. Risk of atrial fibrillation (AF) in patients with obstructive sleep apnea (OSA). Forest plot and funnel plot.” It appears however that the funnel part of this figure was missing probably due to technical error. We added the funnel in figure 3 and corrected the legend. We renumbered accordingly the other figures from figures 3-7 to figures 4-8.
Original: “Forest plot. Risk of atrial fibrillation (AF) in patients with obstructive sleep apnea (OSA). Forest plot and funnel plot.”
Change:
“Risk of atrial fibrillation (AF) in patients with obstructive sleep apnea (OSA). Forest plot and funnel plot.”
- Relation between OSA and AF
U state that age has a relation to OSA, to what extent, may you plese quantify wether there is some cut-off value at which the age is considered higher risk factor than the other mentioned?
Answer:
We understand the question but it was not possible to calculate the cut off because we do not have raw data. However, we were able to calculate the decrease in incidence of AF in OSA patients every 5 years of older age and this was reported in the manuscript in the results section under the header “Relation between OSA and AF” (page 5).
Original: Age (p=0.04) and hypertension (p=0.04) were found to affect the incidence of OSA in AF patients. In contrast, no effect was found for sex (0.44), diabetes mellitus (p=0.62), and BMI (p=0.9).
Changes:
Age (p=0.04) and hypertension (p=0.04) were found to affect the incidence of OSA in AF patients. An increase of 5 years corresponded to a reduction by 9.4% in incidence of AF. In contrast, no effect was found for sex (0.44), diabetes mellitus (p=0.62), and BMI (p=0.9).
- Gender
You stated that no effect was found for sex, fro the antropometric point of view that is surprising. First men have are much more often affected and on the other hand female sex is a risk factor even in the CHADSVASc score. You should mention what was the proportion of men vs. women in the whole cohort and than analyze separately for both genders.
Moreover a baseline characteristics including age, antropometric parameters (BMI, height, weight), prevalence of hypertension, coronary artery disease etc.
In a large number of the papers that were included in the study, there were no detailed data about age and anthropometric parameters. Of course, the articles included these data for the total populations examined but not specifically about the subgroups with and without AF and with and without OSA. In several cases, only the number of patients with and without AF and OSA was reported. We mention in the limitations: “…not all the included studies had detailed data and anthropometric parameters about the characteristics, such as age, sex, BMI, and presence of diabetes or hypertension for all the subgroups of patients (i.e. with and without AF and OSA).”
- Conclusion
The conclusion is completely wrong. You CANNOT assume that treating OSA will reduce the incidence of AF! What data have you shown to draw such a strong conclusion? Indeed the result part accounts for a minority of the article.
Answer:
We changed the conclusion accordingly. We would like to respectfully underline that the main message of the paper was that patients with OSA should be referred early to the cardiologist and this has been included in the conclusions.
Original: Our results show a strong correlation of the existence of OSA with the risk for AF indicating that treatment of OSA may contribute to the reduction of AF recurrence. Older age and hypertension strengthen this correlation. Further research is needed to confirm these findings
Changes: Our results show a strong correlation of the existence of OSA with the risk for AF. Older age and hypertension strengthen this correlation. This suggests that patients with OSA should be referred to a cardiologist for a strict follow up including rest ECG and holter ECG. Further research is needed for the definition of the mechanisms of association between AF and OSA.
Reviewer 3 Report
The authors designed a nice study to investigate the correlation of OSA and AF. OSA significantly increases the risk of AF occurrence and thus needs to be addressed as a strong risk factor for AF recurrence.
The manuscript is well written and designed. Methods are clearly expressed and results are coherent with the methodology. Similarly, conclusions reinforces the main message of the study.
Author Response
The authors designed a nice study to investigate the correlation of OSA and AF. OSA significantly increases the risk of AF occurrence and thus needs to be addressed as a strong risk factor for AF recurrence.
The manuscript is well written and designed. Methods are clearly expressed and results are coherent with the methodology. Similarly, conclusions reinforces the main message of the study.
Answer:
We thank the reviewer for their nice review and comments.
Changes: None requested
Reviewer 4 Report
Review of the manuscript jcm-1556740 entitled “Obstructive Sleep Apnea and Atrial Fibrillation” by Moula et al.
The Authors performed a meta-analysis of published data regarding links between atrial fibrillation and sleep apnea syndrome. The study met the methodical requirements for reviews and meta-analyses. No conflict of interest has been declared.
Overall,
Specific comments
1/ In the Abstract the Authors wrote that the “Obstructive sleep apnea (OSA) is a chronic sleep disorder”. Truly, the OSA is not a sleep disorder. It is a regulatory respiratory abnormality related to the complete or partial collapse of the upper airway during sleep.
Please explain what does mean the word “ dramatically”. Emotional descriptor should be replaced by information what actually the increase is (or was) in certain period or should be omitted.
The phrase “significantly underdiagnosed” is imprecise. Please explain what are limits for “significantly underdiagnosed” as opposed to a “non-significantly underdiagnosed”. It would be better to write “underdiagnosed” only.
2/ In the introduction the Authors properly indicated the clinical significance of AF and OSA. They cited a study of Almeneessier et al. [a case-control study in 498 patients (394 OSA patients and 104 non-OSA patients (controls) who underwent level I attended overnight polysomno-graphy] as background of the statement that patients with OSA have a higher incidence of AF than the general population. I would be a little bit careful. The incidence of OSA was considered by the Authors to be insufficiently estimated in previous studies taking into account a paper of the study of Young et al from 1997 year. They decided to use historical works (which insufficiently estimated OSA incidence???) together to increase the chance of a clearer assessment of the mutual dependence between AF and OSA and its relation to shared comorbidities. It would be properly to use also recent information whilst more data are accessible for next 25 years, I suppose.
3/ Selection strategy is a key point in the presented study. Use of search items like atrial fibrillation (AF) or sleep apnea (SA) did allow to include any (all?) reports from the PubMed database. Actually, ¼ of the included reports (15 out of 20 qualified) have come from the references or by conducting a free search. I wonder if a more precise definition of the items sought could have worked better than used by the Authors. If the Authors had used the term "obstructive sleep apnea" instead of "sleep apnea" the selection would fit the goal more effective. Also searching for the items among “congress’ proceedings from major cardiothoracic and cardiology societies meetings” should not be used as most presentations did not undergo a peer review procedure. Additionally, studies including at least 10 patients are still – in fact - case reports, whilst being not numerous.
Selection strategy used resulted in inclusion of studies with various types of the apnea syndrome (i.e. obstructive, central or both, CPAP use, inconsistent definitions) and atrial fibrillation (runs, paroxysmal, persistent, perpetual, spontaneous, nocturnal, inducible, post-procedural, requiring intervention) as well various conditions for AF/OSA examinations (cardiomyopathies, hypertension, obesity, cardiac surgery, community-based), detection modalities (for AF routine ECG, Holter ECG, nocturnal ECG (one channel)) or follow-up (present at entry, new-onset, incident in future) or follow-up period (none, 5-9 years). In some cases mild OSA were omitted (Tangawa 2006), whilst in other included (Tung 2017). Prevalence of AF or OSA was extremely different in the selected studies. In some papers cardiac arrhythmia (any) has been as the AF equivalent (Gali et al. 2020), whilst AF was proportionally least frequent one. Unfortunately, the Authors included all arrhythmic events.
Day-to-day variability is a weak point in diagnostics of either the FA runs or the OSA events. It should be considered and adjusted for (if possible), or at least considered. It is a source of underdiagnosing of the mentioned clinical conditions.
In many studies, sleep-disordered breathing syndrome is frequently used, whilst the exact feature can be diagnosed by using polysomnographic methods. Also, a common incidence of the central and obturation sleep apneas periods makes the general observation invalid.
In some papers considered only male patients had been included. Does the sex make a bias?
In my opinion, the search strategy applied by the Authors is the most weak chain in the methodical and subsequent statistical procedures that result in conclusions given.
- The shortcomings of the paper has come also from ignoring several meta-analyses performed in the past. The Authors should cite and discuss of the limitations owing the previous analyses. Also other studies that include plenty of subjects (4,082 subjects with the diagnosis of SDB were selected as the study group, and the 575,439 subjects constituted the control group; Chao TF et al. Circ J 2014; 78: 2182–2187) should certainly be included and considered.
- Conclusions. The Authors wrote that “..Our results show a strong correlation of the existence of OSA with the risk for AF indicating that treatment of OSA may contribute to the reduction of AF recurrence..”. Such a statement is clearly surprising as proofs given are hardly to be finds in the results (not in the discussion).
Author Response
The Authors performed a meta-analysis of published data regarding links between atrial fibrillation and sleep apnea syndrome. The study met the methodical requirements for reviews and meta-analyses. No conflict of interest has been declared.
Overall,
Specific comments
- In the Abstract the Authors wrote that the “Obstructive sleep apnea (OSA) is a chronic sleep disorder”. Truly, the OSA is not a sleep disorder. It is a regulatory respiratory abnormality related to the complete or partial collapse of the upper airway during sleep.
Answer:
Dear Reviewer: Thank you for your insightful comments.
There are several mentions in the literature about sleep apnea being a sleep or sleep-related disorder (https://pubmed.ncbi.nlm.nih.gov/35004785/, https://pubmed.ncbi.nlm.nih.gov/21654148/, https://pubmed.ncbi.nlm.nih.gov/24145590/)
However, we agree that this is not the most accurate definition. For this reason, we changed this accordingly.
Original: Obstructive sleep apnea (OSA) is a chronic sleep disorder more common in older men.
Changes:
Obstructive sleep apnea (OSA) is a regulatory respiratory disorder of partial or complete collapse of the upper airways during sleep leading to recurrent pauses in breathing. OSA is more common in older men.
- Please explain what does mean the word “ dramatically”. Emotional descriptor should be replaced by information what actually the increase is (or was) in certain period or should be omitted.
Answer:
Dear Reviewer, thank you for your comment. We agree with your comment so we have changed the word “dramatically” with “significantly”.
- The phrase “significantly underdiagnosed” is imprecise. Please explain what are limits for “significantly underdiagnosed” as opposed to a “non-significantly underdiagnosed”. It would be better to write “underdiagnosed” only.
Answer:
Dear Reviewer, thank you for your comment. We agree with this comment and have changed the phrase.
Original: Nonetheless, the prevalence of OSA in patients with AF is unknown because OSA is significantly underdiagnosed
Changes:
Nonetheless, the prevalence of OSA in patients with AF is unknown because OSA is underdiagnosed
- In the introduction the Authors properly indicated the clinical significance of AF and OSA. They cited a study of Almeneessier et al. [a case-control study in 498 patients (394 OSA patients and 104 non-OSA patients (controls) who underwent level I attended overnight polysomno-graphy]as background of the statement that patients with OSA have a higher incidence of AF than the general population. I would be a little bit careful.
Answer:
Dear Reviewer, thank you for your comments. We agree with this comment. The following (already existing in our review) references support this phrase, i.e. the fact that patients with OSA have increased risk for AF. Huang et al, 2020 (reference 23) “The combined prevalence of OSA in AF patients has been estimated at 21% to 74%”. Perger et al, 2019 (reference 25): “These consequences of OSA …contribute, in turn, to increased cardiovascular risk and, in particular, to the development of chronic systemic arterial hypertension and arrhythmias, especially atrial fibrillation (AF)”.
We have changed this accordingly.
Original: Patients with OSA have a higher incidence of AF than the general population30.
Changes:
Patients with OSA have a higher incidence of AF than the general population23,25,30.
- The incidence of OSA was considered by the Authors to be insufficiently estimated in previous studies taking into account a paper of the study of Young et al from 1997 year. They decided to use historical works (which insufficiently estimated OSA incidence???) together to increase the chance of a clearer assessment of the mutual dependence between AF and OSA and its relation to shared comorbidities. It would be properly to use also recent information whilst more data are accessible for next 25 years, I suppose.
Answer:
Dear Reviewer, thank you for your comments. We agree with this comment and there are newer studies that also indicate the underdiagnosis of OSA. We added one additional reference that supports this statement (Delesie, M., Knaepen, L., Verbraecken, J., Weytjens, K., Dendale, P., Heidbuchel, H., & Desteghe, L. (2021). Cardiorespiratory Polygraphy for Detection of Obstructive Sleep Apnea in Patients With Atrial Fibrillation. Frontiers in cardiovascular medicine, 8, 758548. https://doi.org/10.3389/fcvm.2021.758548).
We have changed this accordingly.
Original: Nonetheless, the prevalence of OSA in patients with AF is unknown because OSA is significantly underdiagnosed31.
Changes:
Nonetheless, the prevalence of OSA in patients with AF is unknown because OSA is underdiagnosed19,20.
- Selection strategy is a key point in the presented study. Use of search items like atrial fibrillation (AF) or sleep apnea (SA) did allow to include any (all?) reports from the PubMed database. Actually, ¼ of the included reports (15 out of 20 qualified) have come from the references or by conducting a free search. I wonder if a more precise definition of the items sought could have worked better than used by the Authors. If the Authors had used the term "obstructive sleep apnea" instead of "sleep apnea" the selection would fit the goal more effective. Also searching for the items among “congress’ proceedings from major cardiothoracic and cardiology societies meetings” should not be used as most presentations did not undergo a peer review procedure.
Answer:
Dear reviewer, thank you for your insightful comments. While constructing the search strategy we used a more specific search. However, when compared with the results of a “wider” search the results of the search that was more specific were less in both quality and quantity so, we decided to choose the broader search in order to not miss any important articles.
In order to confirm this, we ran the search again. On February 8, 2022 we ran the query with “obstructive sleep apnea” instead of “sleep apnea” as suggested and the new query resulted in less results. The new query “("Sleep Apnea Syndromes"[Mesh] OR "obstructive sleep apnea") AND ("atrial fibrillation"[Mesh] OR "atrial fibrillation")” produced 806 results. The original query “("Sleep Original query: Apnea Syndromes"[Mesh] OR "sleep apnea") AND ("atrial fibrillation"[Mesh] OR "atrial fibrillation")” produced 923 results.
Although we did search in proceedings, in order not to omit any relevant data, finally no data from congress proceedings and cardiothoracic meetings were used in the present study. Thus, as no data from proceedings were used, we removed that expression.
Original: An unrestricted literature search was performed using PubMed, Web of Science and Google Scholar Databases, as well as congress proceedings from major cardiotho-racic and cardiology societies meetings.
Changes:
An unrestricted literature search was performed using PubMed, Web of Science and Google Scholar Databases.
- Additionally, studies including at least 10 patients are still – in fact - case reports, whilst being not numerous.
Answer:
Dear reviewer, thank you for your comments. The definition of what a case study is varies. In our review we considered case studies to include only 1 patient. (source: Gopikrishna V. A report on case reports. J Conserv Dent. 2010;13(4):265-271. doi:10.4103/0972-0707.73375 https://www.ncbi.nlm.nih.gov/pmc/articles/PMC3010033/).
We applied the criterion of studies with at least 10 patients in order not to omit any relevant results. In our analysis however, the smallest study, in terms of patient number, included 85 patients. So, regardless of the exact definition of a case study, there were no studies with such a small number of subjects that were included.
- Selection strategy used resulted in inclusion of studies with various types of the apnea syndrome (i.e. obstructive, central or both, CPAP use, inconsistent definitions) and atrial fibrillation (runs, paroxysmal, persistent, perpetual, spontaneous, nocturnal, inducible, post-procedural, requiring intervention) as well various conditions for AF/OSA examinations (cardiomyopathies, hypertension, obesity, cardiac surgery, community-based), detection modalities (for AF routine ECG, Holter ECG, nocturnal ECG (one channel)) or follow-up (present at entry, new-onset, incident in future) or follow-up period (none, 5-9 years). In some cases mild OSA were omitted (Tangawa 2006), whilst in other included (Tung 2017). Prevalence of AF or OSA was extremely different in the selected studies. In some papers cardiac arrhythmia (any) has been as the AF equivalent (Gali et al. 2020), whilst AF was proportionally least frequent one. Unfortunately, the Authors included all arrhythmic events.
Dear reviewer, thank you for your insightful comment. We agree with this comment. These are some of the limitations of the study we conducted and have already been partially addressed in the section of the limitations. We changed the limitations accordingly as follows:
Original: “First, the definition of OSA was not universal and there were partial differences between the studies. Some …”
Change:
“First, the definitions of AF and OSA and the respective diagnostic methods were not universal and there were partial differences between the studies. Some studies…”
- Day-to-day variability is a weak point in diagnostics of either the FA runs or the OSA events. It should be considered and adjusted for (if possible), or at least considered. It is a source of underdiagnosing of the mentioned clinical conditions.
Answer:
The authors agree with this reviewer and in the past they have used alternative meters to study the burden of AF since they believe that standard estimators like Kaplan–Meier estimator cannot be applied in events that cannot meet the hazard assumption. Unfortunately, all current articles in the literature employ these methods and there is no chance to correct in our analysis this bias coming from the original papers.
Changes
We added to the limitations: “…. Fifth, day-to-day variability is a weak point in diagnostics of either the FA runs or the OSA events. It should be considered and adjusted for since it might be a source of underdiagnosing of the mentioned clinical conditions. Unfortunately, all the AF and OA studies share the limitation of using hazard functions for events that cannot meet the hazard assumption and whose initiation and termination cannot be clearly defined. When the continuous implantable monitoring will be employed in OSA patients with AF more insights can be obtained.
- In many studies, sleep-disordered breathing syndrome is frequently used, whilst the exact feature can be diagnosed by using polysomnographic methods. Also, a common incidence of the central and obturation sleep apneas periods makes the general observation invalid.
Answer:
Dear reviewer, we agree with this comment. It is also written in the limitations of the study.
- In some papers considered only male patients had been included. Does the sex make a bias?
Answer:
No. This was tested and excluded.
- In my opinion, the search strategy applied by the Authors is the most weak chain in the methodical and subsequent statistical procedures that result in conclusions given.
Answer:
We already addressed the reviewer’s considerations about the search strategy in our previous answers 5-7.
- The shortcomings of the paper has come also from ignoring several meta-analyses performed in the past. The Authors should cite and discuss of the limitations owing the previous analyses.
Answer:
Dear reviewer, thank you for your comment. We are aware of some meta-analyses on the general subject which however are focusing on different aspects. There is only one meta-analysis (Youssef et al) with the same subject and this was referenced accordingly.
Change:
Previous meta-analyses are focusing mainly in the effect of treatment of OSA on the recurrence of AF (references 1-8). In a recent meta-analysis (Youssef et al, 2018) including 9 studies, a strong association between OSA/sleep disorder is reported.
- Congrete, S.; Bintvihok, M.; Thongprayoon, C.; Bathini, T.; Boonpheng, B.; Sharma, K.; Chokesuwattanaskul, R.; Srivali, N.; Tanawuttiwat, T.; Cheungpasitporn, W. Effect of obstructive sleep apnea and its treatment of atrial fibrillation recurrence after radiofrequency catheter ablation: A meta-analysis. J Evid Based Med 2018, 11, 145-151
- Deng, F.; Raza, A.; Guo, J. Treating obstructive sleep apnea with continuous positive airway pressure reduces risk of recurrent atrial fibrillation after catheter ablation: a meta-analysis. Sleep Med 2018, 46, 5-11
- Labarca, G.; Dreyse, J.; Drake, L.; Jorquera, J.; Barbe, F. Efficacy of continuous positive airway pressure (CPAP) in the prevention of cardiovascular events in patients with obstructive sleep apnea: Systematic review and meta-analysis. Sleep Med Rev 2020, 52, 101312
- Li, X.; Zhou, X.; Xu, X.; Dai, J.; Chen, C.; Ma, L.; Li, J.; Mao, W.; Zhu, M. Effects of continuous positive airway pressure treatment in obstructive sleep apnea patients with atrial fibrillation: A meta-analysis. Medicine (Baltimore) 2021, 100, e25438
- Ng, C.Y.; Liu, T.; Shehata, M.; Stevens, S.; Chugh, S.S.; Wang, X. Meta-analysis of obstructive sleep apnea as predictor of atrial fibrillation recurrence after catheter ablation. Am J Cardiol 2011, 108, 47-51
- Qureshi, W.T.; Nasir, U.B.; Alqalyoobi, S.; O'Neal, W.T.; Mawri, S.; Sabbagh, S.; Soliman, E.Z.; Al-Mallah, M.H. Meta-Analysis of Continuous Positive Airway Pressure as a Therapy of Atrial Fibrillation in Obstructive Sleep Apnea. Am J Cardiol 2015, 116, 1767-1773
- Shukla, A.; Aizer, A.; Holmes, D.; Fowler, S.; Park, D.S.; Bernstein, S.; Bernstein, N.; Chinitz, L. Effect of Obstructive Sleep Apnea Treatment on Atrial Fibrillation Recurrence: A Meta-Analysis. JACC Clin Electrophysiol 2015, 1, 41-51
- Yang, Y.; Ning, Y.; Wen, W.; Jia, Y.; Chen, X.; Huang, M.; Sara, J.D.; Qin, Y.; Fang, F.; Zhang, H.; et al. CPAP is associated with decreased risk of AF recurrence in patients with OSA, especially those younger and slimmer: a meta-analysis. J Interv Card Electrophysiol 2020, 58, 369-379
- Also other studies that include plenty of subjects (4,082subjects with the diagnosis of SDB were selected as the study group, and the 575,439 subjects constituted the control group; Chao TF et al. Circ J 2014; 78: 2182–2187) should certainly be included and considered.
Answer:
Dear reviewer, thank you for your comment and the suggestion of the study by Chao et al. This is a conclusion like ours, but it did not take into account other factors, identifying those that strengthen this association. In other words, other reviewers failed to answer to the question whether this association was independent. From the statistical point of view it was biasing to include results of a metanalysis with overlapping studies.
We discussed it into the paper and added the reference
Changes:
In the discussion we added:
“…This confirms the results from Chao et al. including 57,9521 patients, of which 4,082 had OSA. After a 9,2-year average follow-up, the incidence of AF was 0.7% in patients without OSA and 1.38% in patients with OSA (p<0.001)(62). Nonetheless, OSA and AF share many common risk factors therefore, the presence of one may promote the development of the other. Hence it is not clear whether this association is primary or mediated by shared risk factors. For this reason, we wanted to go further, carrying out a meta-regression to test whether this association was independent or it was the results of the interaction of OSA with other shared cardiovascular risks with common pathophysiological mechanisms”.
- The Authors wrote that “..Our results show a strong correlation of the existence of OSA with the risk for AF indicating that treatment of OSA may contribute to the reduction of AF recurrence..”. Such a statement is clearly surprising as proofs given are hardly to be finds in the results (not in the discussion).
Answer:
Dear reviewer, thank you for your insightful comment. We agree with this correction and so the conclusion was changed accordingly.
Original: Our results show a strong correlation of the existence of OSA with the risk for AF indicating that treatment of OSA may contribute to the reduction of AF recurrence. Older age and hypertension strengthen this correlation. Further research is needed to confirm these findings.
Change:
Our results show a strong correlation of the existence of OSA with the risk for AF. Older age and hypertension strengthen this correlation. This suggests that patients with OSA should be referred to a cardiologist for a strict follow up including rest ECG and holter ECG. Further research is needed for the definition of the mechanisms of as-sociation betwee
Round 2
Reviewer 4 Report
Dear Authors,
The Authors made numerous corrections to the text. The discussion has been supplemented with comments on work limitations. Several references have also been added.
I am aware that it is difficult to change the scope of the research and the methodology. However, since some differences turned out to be marginally significant (i.e. p = 0.049 Fig. 5), it would be advisable to re-review in detail the studies in which patients with central apnea and in which supraventricular arrhythmias other than atrial fibrillation were included. Also, due to the large dispersion of patient numbers and age, the relationships could be determined with mathematically transformed data (i.e. logarithmic or logit).
Kind regards
Author Response
The Authors made numerous corrections to the text. The discussion has been supplemented with comments on work limitations. Several references have also been added.
I am aware that it is difficult to change the scope of the research and the methodology. However, since some differences turned out to be marginally significant (i.e. p = 0.049 Fig. 5), it would be advisable to re-review in detail the studies in which patients with central apnea and in which supraventricular arrhythmias other than atrial fibrillation were included. Also, due to the large dispersion of patient numbers and age, the relationships could be determined with mathematically transformed data (i.e. logarithmic or logit).
Answer We gratefully thank this reviewer. As requested, we conducted the statistical evaluation again, for those papers that refer specifically to AF and those that refer to arrhythmias in general and revised the manuscript, particularly figure 2, accordingly.
Changes. Figure 2 was replaced with figure 2 A and B
Page 5 Line 5 :
"The log of the odds ratios (Log OR) for supraventricular arrhythmias and OSA (Figure 2 A) and AF and OSA (Figure 2 B) are shown in Figure 2 (Funnel diagram in Figure 3). There was no significant difference between those papers that refer to supraventricular arrhythmias and those that refer to AF (0.63 [0.44, 0.83] vs 0.68 [0.48, 0.89], respectively).
was added
Furthermore, the figure is logarithmic as requested